# Mast Cell Interaction with Foxp3^+^ Regulatory T Cells Occur in the Dermis after Initiation of IgE-Mediated Cutaneous Anaphylaxis

**DOI:** 10.3390/cells11193055

**Published:** 2022-09-29

**Authors:** Rasha Msallam, Bernard Malissen, Pierre Launay, Ulrich Blank, Gregory Gautier, Jean Davoust

**Affiliations:** 1Institut Necker Enfants Malades, Centre National de la Recherche Scientifique UMR 8253, Université Paris Cité, Institute National de la Santé et de la Recherche Médicale U1151, 75020 Paris, France; 2Centre d’Immunophénomique, Aix Marseille Université, INSERM, CNRS, 13288 Marseille, France; 3Centre d’Immunologie de Marseille-Luminy, Aix Marseille Université, INSERM, CNRS, 13288 Marseille, France; 4Laboratoire d’Excellence Inflamex, Centre de Recherche sur l’Inflammation, INSERM UMR1149, CNRS EMR8252, Université Paris Cité, 75018 Paris, France; 5UVSQ, INSERM, END-ICAP, Université Paris-Saclay, 78000 Versailles, France

**Keywords:** skin inflammation, mast cells, regulatory T cells, Langerhans cells

## Abstract

Mast cells (MCs) are well-known for their role in IgE-mediated cutaneous anaphylactic responses, but their regulatory functions in the skin are still under intense scrutiny. Using a Red MC and Basophil reporter (RMB) mouse allowing red fluorescent detection and diphtheria toxin mediated depletion of MCs, we investigated the interaction of MCs, Foxp3^+^ regulatory T lymphocytes (Tregs) and Langerhans cells (LCs) during passive cutaneous anaphylaxis (PCA) responses. Using intravital imaging we show that MCs are sessile at homeostasis and during PCA. Breeding RMB mice with Langerin-eGFP mice revealed that dermal MCs do not interact with epidermal-localized LCs, the latter showing constant sprouting of their dendrites at homeostasis and during PCA. When bred with Foxp3-eGFP mice, we found that, although a few Foxp3^+^ Tregs are present at homeostasis, many Tregs transiently infiltrated the skin during PCA. While their velocity during PCA was not altered, Tregs increased the duration of their contact time with MCs compared to PCA-control mice. Antibody-mediated depletion of Tregs had no effect on the intensity of PCA. Hence, the observed increase in Treg numbers and contact time with MCs, regardless of an effect on the intensity of PCA responses, suggests an anti-inflammatory role dedicated to prevent further MC activation.

## 1. Introduction

The skin is a highly regulated and structured immune barrier tissue in contact with a large number of commensals [1]. Mast cells (MCs) constitute an important population of cutaneous immune effectors [2,3]. Localized in the dermis, MCs play an important role in innate and adaptive immune responses. They secrete a large variety of inflammatory and immunoregulatory compounds in response to infectious agents or noxious products such as venoms [4,5]. They are also prime effectors in the initiation of allergies, a type 2 immune disease characterized by the production of Immunoglobulin E antibodies (IgE) to a special class of antigens also called allergens [6]. They have also emerged as new players in the regulation of inflammatory responses and immunological tolerance in diverse settings, advocating for a reappraisal of their function [7,8,9].

Besides MCs, the skin contains a variety of antigen-presenting cells with defined localizations, subsets and activation/migration properties. They include Langerhans cells (LCs) being predominant in the epidermis and various dendritic cell (DC) subsets in the dermal layer such as conventional DC type 1 (cDC1), type 2 (cDC2) and the recently defined human type 3 (cDC3) with a role in psoriasis as well as resident and infiltrating macrophages [10,11,12,13,14,15,16]. Furthermore, the compartment of Foxp3^+^ regulatory T lymphocytes (Tregs) found in the skin has raised considerable attention since they can be educated both in the thymus and at environmental interfaces including in the skin [17,18]. Tregs exert exquisite immunosuppressive activities by interacting with cells of the innate and adaptive immune system. Foxp3 null mutations result in dysfunctional Tregs and cause fulminant fatal autoimmune inflammation, systematically associated with exacerbation of overt immune responses and hyper-IgE syndrome both in humans (IPEX syndrome) and in mice with noticeable skin inflammation [19].

Numerous reports have proposed a crosstalk between MCs and other components of the immune system in the skin such as Tregs [20], DCs [21] and more recently type 2 innate lymphoid cells (ILC2s) in the dermis [22,23]. A functional MC-Treg crosstalk has been shown in a skin graft model under tolerizing conditions [24]. Upon grafting, activated MCs released GM-CSF thereby increasing tolerogenic DCs that support Treg functions in draining lymph nodes [24,25]. In turn, Treg-derived interleukin-9 (IL-9) promoted MC recruitment into transplanted allografts contributing to the maintenance of allograft tolerance [24]. The direct contact of MCs with Tregs through OX40-OX40L interactions suppressed IgE-mediated MC degranulation [26,27,28]. In addition, Treg depletion enhanced MC-induced systemic anaphylactic responses [26,29] and food anaphylaxis [30]. By contrast, Tregs can also enhance production of certain cytokines such as interleukin-6 (IL-6) and interleukin-17 (IL-17) by MCs, a feature shown to depend on transforming growth factor-β (TGF-β) [27,30].

We previously developed a knock-in RMB (Red Mast cell and Basophil) mouse model that allows both visualization and conditional ablation of MCs and basophils [31]. Both cell types express the endogenous tandem dimer Tomato (tdT) red fluorescent marker and are coincidently sensitive to diphtheria toxin (DT) mediated conditional ablation. Due to their short half-life basophils are fully replenished after 12 days, whereas repopulation of the long-lived MC population required several months [31].

In this study, using real-time confocal imaging and dedicated reporter mouse models, we explored in vivo the dynamic properties, recovery rate and interactions of MCs with Tregs, and with LCs in the skin occurring under homeostatic conditions and during initiation of an inflammatory passive cutaneous anaphylaxis (PCA) response. Our results show that shortly after initiation of the inflammatory response, while MCs and LCs remain sessile, Tregs promptly accumulate in the dermis. Moreover, our results demonstrate here that during PCA, Tregs establish prolonged contacts with MCs, highlighting the occurrence of a functional crosstalk between MCs and Tregs in the dermis.

## 2. Materials & Methods

### 2.1. Mouse Strains

Mice were housed under specific pathogen-free conditions at the mouse facilities of the Bichat Medical School. Red Mast Cell and Basophil reporter (RMB) mice have been described [31]. They allow dual expression of the tandem dimer Tomato (tdT) red fluorescent protein and the diphtheria toxin (DT) receptor in all MCs and basophils expressing a functional type I high affinity IgE Fc receptor (FcεRI). Indeed, as previously described, in RMB mice the 3′-UTR of the Ms4a2 gene encoding the FcεRIβ chain includes a cassette composed of an internal ribosomal entry site (IRES), a sequence coding for tdT, a 2A cleavage sequence, and the human diphtheria toxin receptor (hDTR) [31]. They were crossed with knock-in *Langerin-IRES-eGFP* (Langerin-eGFP) mice that allow eGFP detection of Langerhans cells localized in the epidermis as described [32]. They were also crossed with *Foxp3-IRES-eGFP* (RMB-Foxp3-eGFP) reporter mice [33] allowing eGFP detection of Tregs. In all experiments, unless specified, mice were used at an age of 10 to 14 weeks. For MC and basophil depletion, mice were injected intraperitoneally (i.p.) twice (two days apart) with 1 µg per mouse of *Diphtheria Toxin* (DT); littermate mice injected with PBS were used as controls. To allow basophil repletion in the absence of significant MC repletion, mice were used 12 days after the second i.p. DT injection as previously described [31]. All animal experiments were performed in accordance with the French Council of Animal Care guidelines and national ethical guidelines. The study was approved by the local ethical committee (comité d’éthique en expérimentation animale, Faculté de Médecine Site Bichat, Université Paris Cité) and by the Department of Research of the French government under the animal study proposal numbers APAFIS# 14682 and 14156.

### 2.2. Passive Cutaneous Anaphylaxis (PCA) Experiments

At day −2 and day 0, mice were injected twice i.p. with either DT (1 µg) or PBS. PCA experiments were then performed as described with slight modifications [34]. Briefly, on day 9 after last DT injection, mice were passively sensitized with mouse anti-DNP IgE (clone H1-ε-26) (100 ng in 50 µL) by intradermal injection in the right ear skin or vehicle (PBS) in the left ear skin as control. On day 12, mice were challenged with 150 µg (in 150 µL) of DNP-HSA (Sigma–Aldrich, Saint Louis, MO, USA) in 1% Evans blue dye (Sigma–Aldrich) by retro-orbital injection. Mice were either sacrificed 30 min after the antigen challenge or used for confocal imaging at indicated times after antigen challenge. For Evans’s blue quantification, each ear was removed and immersed in 200 µL of Formamide for 48 h at 55 °C. Evans’s blue concentration was determined by measurement of OD at 610 nm. In some experiments PCA was performed after anti-CD25 rat IgG1 mAb (PC61) mediated functional ablation and deletion of Tregs [35,36,37]. Briefly, mice were injected i.p. with 0.2 mg per mouse of PC61 mAb or isotype control at day 0. Mice were then passively sensitized at day 2 by intradermal injection with mouse anti-DNP IgE in the right ear skin or vehicle (PBS) in the left ear skin. On day 4, mice were challenged with 150 µg (in 150 µL) of DNP-HSA (Sigma–Aldrich, Saint Louis, MO, USA) in 1% Evans blue dye (Sigma–Aldrich) by retro-orbital injection.

### 2.3. Intravital Confocal Microscopy

All images and videos were analyzed by using Image J (https://imagej.nih.gov/ij/, accessed on 1 August 2022) and 4D image analysis software (Imaris).

Mice preparation: Mice were anesthetized by injecting i.p. 120 µL of the Xylasin/Ketamine cocktail mixed with PBS (1:1:2); any supplementary injection of anesthesia during the confocal experiment were performed intra-muscularly (i.m). Mice were maintained under observation on the microscope stage in a chamber regulated at 28 °C. These very mild anesthesia conditions were applied for less than 2 h.

Static mosaic images and time lapse video acquisitions: We adapted regular intravital confocal imaging as described previously [32] to visualize both the epidermis and the dermis of ear skin in typical non-invasive inspections. For that, we made a trade-off between XY resolution and Z depth of focus and tuned the instrument with a somewhat larger pinhole size than usual (1.5 to 2 time the Airy disk) as previously established for vital ear skin imaging [32]. Static mosaic images and time lapse video acquisitions were performed in the ear-skin using Leica TCS SP5 and SP8 microscopes (Leica Microsystems, Wetzlar, Germany). 405 nm, 488 nm, and 561 nm lasers are used for Blue (autofluorescence), eGFP, and tdT excitation, respectively. Typical confocal recording times were 30 to 60 min to acquire complete dual channel 3D and 4D data sets for static mosaic images and video data sets, respectively. Static mosaic images were recorded by assembling a series of juxtaposed dual channel 3D frames (from 20 to 30 frames) using the mosaic routine allowing automatic control of the XYZ motorized stage of the Leica TCS SP5 and SP8 confocal microscopes with a typical duty cycle of 2 to 3 min between the individual frames. Video confocal image dual channel 4D data sets were recorded by assembling a series of time lapse dual channel 3D frames using the time lapse routine and automatic control of the Z motorized stage of the Leica TCS SP5 and SP8 confocal microscopes, with a typical duty cycle of 4 to 5 min between time points. For each time point, the initial raw X, Y, Z stacks are composed of 20 to 30 individual sections covering the dermal layer. On some occasions, individual sections allowed us to visualize all the details including the presence of nuclei structures, which exclude the fluorescent tdT marker. Next, we processed all raw image acquisition stacks using maximum Z projection of 4D (X, Y, Z, color) data sets acquired for the static mosaic images and dual channel 5D (X, Y, Z, color and time) video acquisitions to visualize on single micrographs all MCs, Tregs and LCs present in different focal planes within the specimens. tdT^+^ MCs were revealed in the red fluorescence channel and a third blue spectrum fluorescence channel imaging was performed to differentiate the tdT^+^ red only MCs from the yellow auto-fluorescent structures such as hair follicles and associated sebaceous glands present at the basal position around each follicle. eGFP was imaged using the green channel. All images were acquired in 1024 × 1024 format with a 40X HCS PLAPO objective. Dual fluorescence channel 4D data sets were collected over the 30–45 μm thickness with 1.5 μm Z step size using the 20X PLAPO objective (field of 450 μm × 450 μm, 1 Mpixel image). Maximum Z projections of 3D stacks were obtained at all time points of multi-dimensional dual channel 4D image stacks yielding time lapse dual channels videos animations of the ear-skin. Typically, for video imaging, mice were monitored for 30 to 60 min with a duty cycle of 2 to 4 min to obtain repetitively dual channels images with full depth 3D imaging, yielding dual color animated XY video images.

### 2.4. Determination of Cell Numbers, Tracking Cellular Motility and Velocity

MC and Treg numbers in the skin were manually quantified in acquired images. After converting the surface of the analyzed images from pixels into µm^2^, the total number of MCs/mm^2^ was calculated by the following formula: $\frac{\;Total\;number\;of\;quantified\;cells}{Real\;surface\;dimension\;\left(µm2\right)}$
*×* 10^6^. To visualize cell motility on static images, we merged three time points (as indicated) of images using Red, Green, and Blue (RGB) color coding for each time point respectively. The resulting images are composed of immobile cells in white, forward movements and projections of the cells in red and backward retractions of the cells in blue. For an improved detection of the motility of cells in 5D videos, we further applied this RGB motility enhancement for all consecutive triad time points within the videos as previously performed [32]. Briefly, each video Z stack was copied in triplicates using ImageJ software, one copy was converted in blue color, one in green color and one in red color. We then merged the 3 videos, but introduced a time shift of minus one time frame for the blue video and plus one time frame for the red video. In the resulting RGB video, each time frame has a color coding of red for time frame +1, green for the actual time frame and blue for time frame −1. Thus, we were able to identify visually the cell motility in 5D videos as previously performed [32]. For cell tracking and quantification of cell displacement velocities, we used the Imaris software yielding overlays with white displacement tracking lines over the micrographs. Quantification of cell velocities in μm per min was then calculated using a tracking plug-in (http://rsbweb.nih.gov/ij/plugins/track/track.html, accessed on 1 August 2022). Of note, to allow for the proper quantification of Treg motions with the software, and due to their high motility, we reduced the duty cycle of the time lapse confocal recording to 40 from 60 s. For that, we reduced the size of the field to 512 × 512 pixels and examined a reduced volume in the dermis with 8 to 12 consecutive sections corresponding to about 20 μm thickness in the dermis.

### 2.5. Quantification of Cell-Cell Interactions

We used correlation maps as previously described to identify contacts between two cells labeled with different fluorophores as previously described [38,39]. Theses maps identify local similarities between two fluorescence profiles from the computation of a correlation coefficient on a local scale (Gaussian window) around each pixel in the images. The local correlation coefficient is either positive for colocalized, superimposed profiles, null for unrelated profiles, or negative for juxtaposed profiles, which is typical at contact sites [38,39]. We computed local correlation coefficient maps of raw images comparing tdT (red channel) and eGFP (green channel) using a Gaussian window of 5 pixels equivalent to a ±3.6 μm diameter to score contact sites within this distance scale. We displayed the negative component in the white channel for each given pixel on the raw image. Independent fields of over 400 × 400 µm surface areas per tissue section were inspected using a numerical aperture 20× objective to visualize Tregs counting a total of 22 Tregs in control experiments and 71 Tregs in PCA experiments. Using the NIH ImageJ software, we extracted the contact points (in white) for each time frame within videos, quantified their occurrence and monitored the duration of each cell-to-cell contact. These results were then plotted considering three categories to assess transient, intermediate and long term contacts.

## 3. Results

### 3.1. Mast Cells Are Sessile and, after Conditional Ablation, Undergo Slow Repopulation Kinetics in the Skin

We used regular intravital confocal imaging to visualize both the epidermis and the dermis of ear skin in typical non-invasive inspections performed on ketamine anesthetized RMB mice maintained under observation at 28 °C. These very mild anesthesia conditions were applied for less than 3 h, allowing successive imaging of the same animal every two days without detectable pathogenic signs. tdT^+^ MCs were detected as described in Material & Methods. On all confocal video sequences, we found that MCs (in red) are sessile, showing an exact superimposition of sequential MC images in the red-green-blue (RGB) color code overlaying images captured at time points 0, 7 and 14 min or in videos captured up to 60 min (Figure 1A, Appendix A). Of importance, on single sections, MC nuclei are clearly visible as tdT negative holes on most MCs visualized with arrows (Figure 1A). To visualize all MCs present in different focal planes in the dermis, we processed all further micrographs with a Z maximum fluorescence projection software, with the drawback of a loss of intracellular details such as nuclei holes (Figure 1B–E). Further figures displaying other eGFP labeled cells such as Tregs and LCs were equally processed with Z maximum projection of 3D and 4D data sets.

Next, we performed large field inspections and confirmed using maximal Z projection images, that MCs form a dense layer of dispersed cells present in the dermis at the steady state (Figure 1B and Appendix A). The distribution of MCs in the dermis is well organized constituting a dense network in agreement with their sentinel function. The apparent circular distribution of MCs around hair follicles is likely related to the vertical disposition of the dermis layer around hair follicles (Figure 1B and Appendix A). To ascertain that MC distribution is restricted to the dermis and cannot contact LCs found in the epidermis, we crossed RMB mice with knock-in Langerin-eGFP mice [32]. High resolution intravital confocal imaging of dual labeled Langerin-eGFP LCs and tdT^+^ MCs confirmed the existence of two individual networks, that of LCs in the epidermis and that of MCs in the dermis (Appendix A).

We then performed DT injections to deplete MCs and monitored the kinetics of their repopulation after conditional ablation (Figure 1C–E). Profound DT-mediated MC ablation was evidenced at day 6 (Figure 1C). Some tdT^+^ MCs appeared at day 11 and a substantial repopulation of tdT^+^ MCs is visible 3 months after DT treatment (Figure 1D,E and Appendix A). Still, the distribution and morphology of repopulating MCs differed from that of steady state MCs, revealing a more uneven distribution and more elongated cells with irregular contours evidenced on large mosaic images (Compare Appendix A). To quantify the time course of MC repopulation, MCs were enumerated on all micrographs collected from the ear skin of RMB mice prior to and after DT infusions up to day 120 (Figure 1F). The initial levels of ~360 MCs per mm^2^ found on planar images were comparable with previous assessments of MC numbers in the skin [40] with a mean MC inter-distance of 53 µm in this network. Second, we found that the kinetics of MC repopulation was slow, reaching about 150 MCs/mm^2^ at 3 months (Figure 1F), comparable to the slow recovery of MCs in the peritoneum reaching about 50% six months after DT treatment in RMB mice [31].

### 3.2. DT-mediated Depletion of MCs Evidences their Role in Passive Cutaneous Anaphylaxis

DT treatment, besides depleting MCs, also depletes basophils. To confirm the reported unique involvement of MCs in the PCA response [34,41], we performed a PCA experiment 12 day after DT injection (Figure 2A). After this period, the basophil compartment is fully replenished as reported [31], whereas MCs remain essentially depleted (Figure 1D,F). The development of PCA in DT-treated mice at day 12 after DT injection was evaluated by measuring Evans blue extravasation in the skin, as shown in the experimental anti-DNP IgE- (left) or control PBS-sensitized (right) ears (Figure 2B). While in the absence of DT treatment (-DT) dye extravasation was clearly visible in the anti-DNP IgE sensitized left ear, DT treatment essentially abrogated extravasation in the left ear 12 days post-DT mediated MC ablation (Figure 2B). The injection of DT alone to eliminate MCs had no inflammatory effects in the local tissue as such, since no difference in Evans blue extravasation was noticed between PBS-sensitized and/or IgE-sensitized ears after DT treatment (Figure 2B). Evans blue extravasations in the skin were further quantified from the ears, clearly supporting the fact that MC ablation nullifies the PCA reaction in the skin (Figure 2C). Furthermore, as the sensitizations were performed after complete reconstitution of the basophil compartment at day 12 post-DT treatment, this experiment clearly demonstrates the unique role of MCs in the skin PCA response.

### 3.3. MCs Interact with Tregs after Initiation of the PCA

As MCs were reported to establish a functional crosstalk with Tregs [26,27,29,30], we wondered whether activated MCs could attract and interact directly with Tregs in the dermis. Hence, we crossed RMB mice harboring tdT red labeled MCs with Foxp3-eGFP^+^ green Treg reporter mice and analyzed the distribution of Foxp3-eGFP^+^ Tregs by live imaging of the ear skin in RMB-Foxp3-eGFP mice. At the steady state (Figure 3A) some Tregs can be detected as small round cells showing a scattered but uneven distribution, albeit they are less abundant than MCs as confirmed by enumeration (Figure 3B). Occasionally Foxp3-eGFP^+^ Tregs can be found to interact with MCs (Figure 3A, arrows). In clear contrast to MCs, Tregs are highly motile under steady state conditions as exemplified on video confocal recordings (Appendix A). After MC ablation, the same type of scattered Treg uneven distribution was found with no apparent relation with repopulating MCs (Figure 3C). When enumerating Tregs, we found that MC ablation had no influence on the number of Tregs present at the examined time-points (Figure 3B).

We next performed real-time imaging of MCs and Tregs in RMB-Foxp3-eGFP mice during an ongoing inflammatory PCA response in the ear skin. Our results show that in control PBS-sensitized ears (from 2 h to 3 h post Ag challenge), merged MC and Treg images (Figure 4A–C and Appendix A) indicate that MCs remain sessile with few Tregs being present as seen above at the steady state (Figure 3A). In contrast to MCs, Tregs were motile establishing occasionally contacts with MCs (Figure 4B, Appendix A). When analyzing Treg behavior after the initiation of PCA (Figure 4D–F, Appendix A and Appendix A) we observed a strong recruitment of Tregs in the dermis when compared to control PBS-sensitized ears at 2 h to 3 h post injection, with an approximate 15-fold increase as evaluated by Treg enumeration (Figure 4G), while at later time points (24 h) their numbers declined again to baseline levels (Figure 4G). After the initiation of PCA in IgE-sensitized mice, Tregs remained highly motile wobbling around MCs, but their mean velocity did not significantly change when compared to control PBS-sensitized ears (Figure 4B,E,H). We further evaluated the duration of MC-Treg contacts using dedicated image analysis software as detailed in Materials & Methods. Importantly, as assessed with image analysis software described previously [38,39], the contact time of Tregs with MCs increased markedly 2 h after PCA challenge with a majority of long-lived contacts lasting up to 50 min or even more following PCA challenge. These long-lived contacts were not observed in control PBS-sensitized ears (Figure 4C,I). Importantly, the majority of contacts established between the newly recruited Tregs with the resident MCs were transient primarily lasting between 10 to 50 min (Figure 4I).

As Tregs have been reported to dampen MC degranulation including in in vivo systemic anaphylaxis [26,29] we next wished to analyze their possible impact on a local skin anaphylactic response upon mAb-mediated ablation of Tregs performed by mAb intraperitoneal injection before intradermal IgE or PBS control sensitization of the skin. The results show that upon PC61 mAb-mediated ablation and deletion of Tregs [35,36,37] the induced inflammatory response, albeit slightly lower, was not significantly different compared to Treg non-depleted control mice (Figure 5). Therefore, Tregs do not significantly control the outcome of the fast PCA response induced by MC degranulation in the dermis (Figure 5).

### 3.4. Langerhans Cells Remain Sessile While Exhibiting Constant Sprouting of Their Dendrites

Finally we also performed real-time imaging of control PBS-sensitized or IgE-sensitized ear skin of RMB × Langerin-eGFP mice 2 h to 3 h post Ag challenge. The data revealed that like MCs (Figure 6A, Appendix A) epidermal Langerhans cells remain sessile both in control PBS-sensitized ears and after the initiation of PCA in IgE-sensitized ears. In contrast to LCs, some motile dermal Langerin+ cells, which primarily comprise cDC1 [42] are observed in the dermis (Appendix A). Close inspection using RGB color code overlay images superimposing time points 0, 7 and 14 min revealed that in contrast to cell bodies, LC dendrites exhibited constant dendrite sprouting. Sprouting was observed both in control PBS-sensitized and IgE-sensitized ears after challenge with Ag as exemplified by the appearance of colored dendrites extremities in RGB mode images (Figure 6B).

## 4. Discussion

MCs are abundant hematopoietic effectors in barrier tissues present particularly at environmental interfaces such as the skin, the gut mucosal barrier and the lungs [3,43]. They play an important role in allergic reactions, but are also increasingly recognized as key regulatory cells in inflammatory processes, tissue regeneration and repair responses [3,43,44,45]. This involves their ability to secrete upon activation, a vast array of inflammatory mediators either stored in cytoplasmic granules or newly synthetized [46,47,48]. While many studies on MC functions have been performed in *c-kit*-dependent models of MC-deficiency, these mice do not allow MC visualization and require MC reconstitution experiments [41,49]. Alternative c-Kit independent MC-deficient mouse models have been engineered, targeting either MC-specific proteases or IL-4 intronic enhancers [41] or the FcεRIβ chain as the case for the RMB mice [31].

The RMB mouse model offers several advantages: (i) MCs express the tdT reporter allowing unprecedented dual channel real-time imaging with other eGFP reporters to unravel MC crosstalk with other resident immune cells in the skin, such as Langerin-eGFP^+^ DC or Foxp3-eGFP^+^ Tregs here. (ii) DT-based MC conditional ablation leads to sudden and profound MC removal in mice with a normally developed hematopoietic system enabling deletion. (iii) The vastly different repopulation kinetics of MCs and basophils, which are both targeted in the present FcεRIβ knock-in mouse allows one to differentiate the role of slowly repopulating MCs from that of the rapidly repopulating basophils [31]. Using this RMB mouse model, we showed the role of MCs during septic shock in the peritoneum and found that peritoneal MCs inhibit macrophage phagocytic functions through release of pre-stored IL-4 within 15 min after bacterial encounter, thereby impeding the resolution of an acute septic shock in the gut [31]. We also found that the RMB model offers a unique possibility to visualize, track and deplete conditionally the MCs in settings of kidney inflammation [50] and skin transplantation [40]. In addition, being able to deplete MCs during the onset of the inflammatory process in the kidney, we found that their action during the early phase is crucial for their inflammatory contribution to the disease [50].

Our primary aim in this study was to visualize the effect of MC activation in vivo during an IgE- and MC-induced PCA response on the skin regulatory T cell compartment. Confirming previous results obtained with other MC deficient models [34,41], we show here that conditional ablation of MCs reduced the IgE-dependent PCA reaction to control levels. Our results obtained in a *cKit*-independent MC ablation model demonstrate the essential role of MCs in this skin inflammatory reaction.

Previously, OX40L/OX40 MC-Treg interactions were shown to empower Tregs to inhibit MC degranulation in vitro [26] and in vivo in a model of passive systemic anaphylaxis [26,29], as well as in an active model of food anaphylaxis [30]. Furthermore, in vitro studies clearly revealed the ability of Tregs to block IgE-mediated MC degranulation both in murine and human co-cultures. This anti-inflammatory activity was abolished when Tregs were deficient for OX40 or when the interaction with the MC-expressed OX40L was blocked [26,27]. Additional functional studies demonstrated that the direct contact between Tregs or soluble OX40 and MCs resulted in the inhibition of calcium mobilization and MC degranulation, but did not inhibit cytokine secretion [20,27,28]. Concerning the latter, it was shown that Tregs activate MCs to produce IL-6 and IL-17 through surface bound TGF-β thereby skewing in return Tregs and effector T cells into IL-17-producing T cells [27,30]. However, the dynamics of MC-Treg interactions between both cell types have not been investigated in an in vivo context in an IgE- and MC-driven disease model. Hence, using RMB-Foxp3-eGFP mice, we visualized and tracked in vivo inhibitory MC-Treg interactions both at homeostasis and after PCA in the ear skin. Our results showed that few Tregs were present at the steady state as well as in control PBS-sensitized, Ag challenge ears. Under these conditions, Tregs showed an uneven distribution and represented in the skin only a minor fraction (about 20-fold less) as compared to MCs. These Tregs were motile, establishing from time to time contacts with MCs. These contacts were majorly short-lived lasting less than 10 min, albeit longer contacts (10–50 min) could also occasionally be detected. The situation changed dramatically within 2 h to 3 h after local IgE/Ag-mediated MC degranulation as we observed an important infiltration of Tregs to the inflammatory site, increasing their numbers by about tenfold. This increase was, however, of transient nature as their numbers essentially returned to baseline levels at 24 h after challenge. In addition, at 2 h to 3 h after initiation of the PCA response the contacts between Tregs and MCs became more frequent with an increased duration. The majority of these durations were in the 10 to 50 min range and a significant number of contacts exceeded 50 min. Although there was no apparent change in Treg velocity, it seems possible that the more frequent and prolonged contacts somewhat slowed down their measured velocities after PCA and led to Treg wobbling around MCs. Together, our data clearly revealed that, while no significant changes were seen in MC-sufficient and MC-depleted animals at homeostasis, Ag challenge induces a prompt MC-induced response of the Treg compartment resulting in a massive Treg infiltration and frequent contacts with MCs. The observed increase in Treg numbers could be the direct result of released histamine by MCs. Indeed, it was previously shown that Tregs migrate to an inflammatory site via activation through their H4 receptor [51]. Whether this also increases their adhesiveness remains to be investigated. However, in our study, functional antibody-mediated ablation of Tregs did not significantly reduce the early phase of the PCA response monitored through Evans blue dye extravasation. This contrasts with previous in vitro experiments and in vivo data of passive systemic anaphylaxis and active food anaphylaxis [26,29,30]. It is likely that in the skin microenvironment, the few Tregs present at the time of the initial challenge are not sufficient to significantly dampen the MC response. The huge and rapid increase in Treg numbers at later time points supports the view that a major task of the skin infiltrating Tregs is to prevent further MC degranulation and ensuing inflammatory responses. The described ability of Tregs to activate MCs to release certain cytokines such as IL-6 and IL-17 after contact [30] may in addition skew the response to enhance protective inflammatory actions. Our data also show that Treg action is clearly of transient nature as in the absence of additional stimulation, Tregs returned back to baseline levels 24 h after the initial challenge as observed in other inflammatory models [52]. This suggest that Treg function is to transiently dampen any further inflammatory response and MC degranulation.

In addition to Tregs, we also analyzed whether MC activation affected possible contacts of MCs with LCs or whether MCs were able to trigger a change in behavior of epidermal LCs. Our data clearly show that LCs are located in a distinct layer of the skin corresponding to the epidermis and are not in contact with dermal MCs at homeostasis or after MC degranulation. Thus, LCs are less sensitive to skin epithelial signals generated by MCs and remain sessile during a short-term PCA response, contrasting with the migratory profile of LCs found after tape stripping [32]. Indeed, our motility analysis in PBS-sensitized, and during PCA in IgE-sensitized animals showed that LCs, like MCs, remain sessile exhibiting constant sprouting behavior with no apparent change in activity after the initiation of PCA.

In conclusion, we have provided here direct visual evidence for the role of MCs in Treg attraction and interaction in a MC-induced inflammatory response in the skin. While the presence of Tregs in low abundance did not seem to impact significantly the strength of the initial skin inflammatory response, our data are consistent a role of Tregs in the later phase of PCA [26,29]. We would like to suggest that in this context, pre-activated Tregs have a preventive action, blocking any further MC degranulation events. Of note, Tregs could also be attracted into the dermis by inflammatory signals generated by immune cells other than MCs, resulting in the attraction of Tregs empowered to control MC activation. In contrast with Tregs, no apparent crosstalk became evident between LCs and MCs, supporting the fact that LCs and MCs have sentinel activities, which are independent from each other in the epidermis and the dermis. Our results revealed that crosstalk between MCs and Tregs occur through direct contact in the inflammatory skin environment, supporting the possibility that Treg-MC interactions play an important role to control later phases of the inflammatory response. Unraveling MC and Treg interactions and molecular pathways involved, as well as functional consequences of these interactions, may provide important new insights and treatment modalities for multiple skin immune disorders.

## Figures and Tables

**Figure 1 cells-11-03055-f001:**
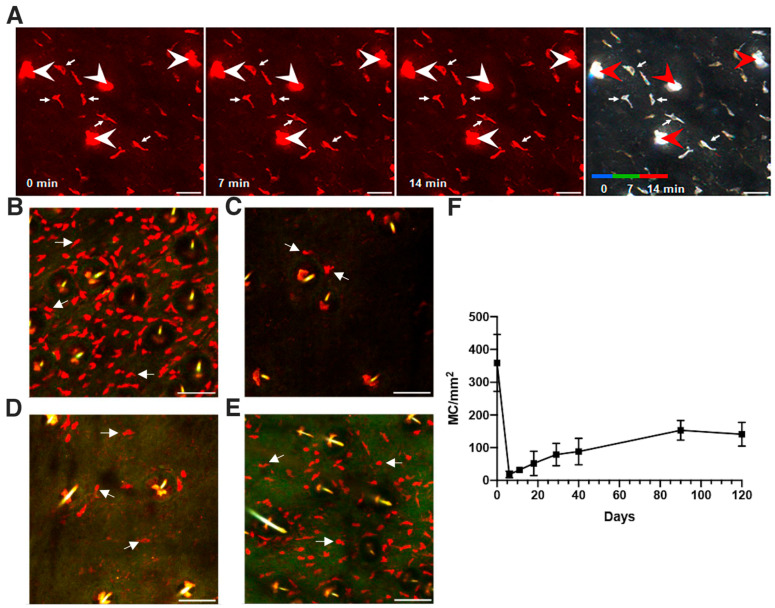
Mast cells are sessile and are slowly replenished after conditional ablation in the ear skin of RMB mice. (**A**) Visualization of tdT^+^ MCs and their dynamics in the ear skin of RMB mice by intravital confocal microscopy. Single optical sections of 1.5 μm thickness are represented choosing a mid-position with maximal number of MCs in the dermis, and were obtained at indicated time points from the 4D image stacks of the ear-skin monitored here for 14 min. Hair follicles and associated sebaceous glands are pointed with large arrowheads, whereas MCs pointed with arrows appear as thin elongated objects. Three sequential time points (0 min, 7 min and 14 min) were merged in an overlay RGB image on the right panel. The resulting white color indicate that MCs are sessile over this time frame. (**B**–**E**) Maximum Z projections of 3D stacks (20–30 sections over a total of 30–45 µm thickness) were performed to visualize the kinetics of repopulation of tdT^+^ MCs in the ear skin after conditional ablation following DT i.p. injection. (**B**) Before DT injection, tdT^+^ MCs form an array of mostly elongated cells at the steady state. (**C**) 6 days after DT injection, very few tdT^+^ MCs are present, while (**D**) sparse tdT^+^ MCs are found 11 days post DT injections. (**E**) Numerous tdT^+^ MCs are found 3 months after conditional ablation of MCs. (**F**) Time course of MC repopulation after conditional ablation. MCs were enumerated in the skin of RMB mice prior (day 0) and after indicated times of DT treatment. Each quantification was performed on Z projection images as in B-E in at least three independent fields of over 400 × 400 µm from two to four mice and expressed as total MC numbers per mm^2^. Arrows in (**B**–**E**) point to MCs. Scale bars (**A**) 30 µm and (**B**–**E**) 100 µm.

**Figure 2 cells-11-03055-f002:**
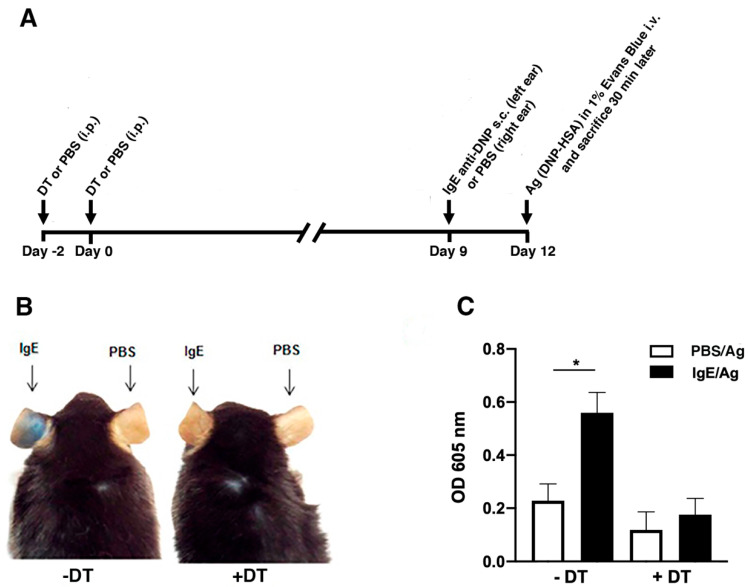
Passive cutaneous anaphylaxis reaction is impaired after conditional ablation of mast cells. (**A**) Schematic diagram of the protocol of the PCA experiment as performed in B. RMB mice were treated or not with 2 DT injections i.p. (1 µg) at day −2 and day 0. Then mice were passively sensitized s.c. with 100 ng anti-DNP IgE (left ear) or with PBS (right ear) at day 9. Three days later, mice were challenged i.v. with Ag (DNP-HSA, 150 μg) in 1% Evans blue in PBS (final volume 150 µL) and mice were sacrificed 30 min later. (**B**) Photomicrographs of dye extravasations of representative mice submitted to a PCA experiment according to the protocol depicted in A. IgE-sensitized ears of non DT treated (−DT) mice exhibited dye extravasation in the skin after Ag challenge, while DT treated (+DT) mice (MC-depleted and basophil repleted at day 12) had only minimal signs of extravasation in IgE-sensitized ears, comparable to PBS-sensitized ears of (−DT) and (+DT) mice. (**C**) Quantification of Evans blue extracted from ears of mice in corresponding experiments is shown. Number of mice correspond to 4 mice per condition. Statistical analysis was based on an unpaired *t*-test (* *p* < 0.05) between groups.

**Figure 3 cells-11-03055-f003:**
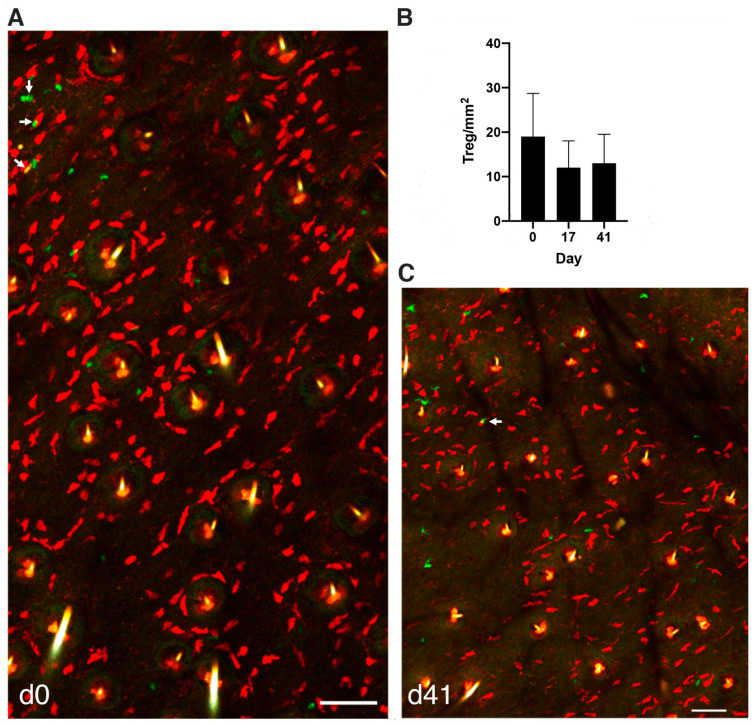
Distribution of Foxp3^+^ Tregs in the skin is unaffected by conditional ablation of mast cells. (**A**) Dual visualization of Foxp3-eGFP^+^ Tregs and tdT^+^ MCs in the ear skin of RMB-Foxp3-eGFP mouse at steady state before DT injection (d0) on mosaic images of maximum Z projection of 13 sections (Z step size of 1.5 µm). Arrows indicate motile Tregs that are in contact with sessile MCs. (**B**) Quantification of Tregs at steady state and after depletion of MCs at indicated days. Each quantification was performed in at least three independent fields of over 400 × 400 µm Z projection images from two to four mice and are expressed as total numbers of Tregs per mm^2^. (**C**) Dual visualization of Foxp3-eGFP^+^ Tregs and tdT^+^ MCs in the ear skin of RMB-Foxp3-eGFP mice at day 41 (d41) after DT injection on mosaic images of maximum Z projection of 13 sections (Z step size of 1.5 µm). The arrow indicates a motile Treg in contact with a sessile MC. Scale bars (**A**,**C**) 100 µm.

**Figure 4 cells-11-03055-f004:**
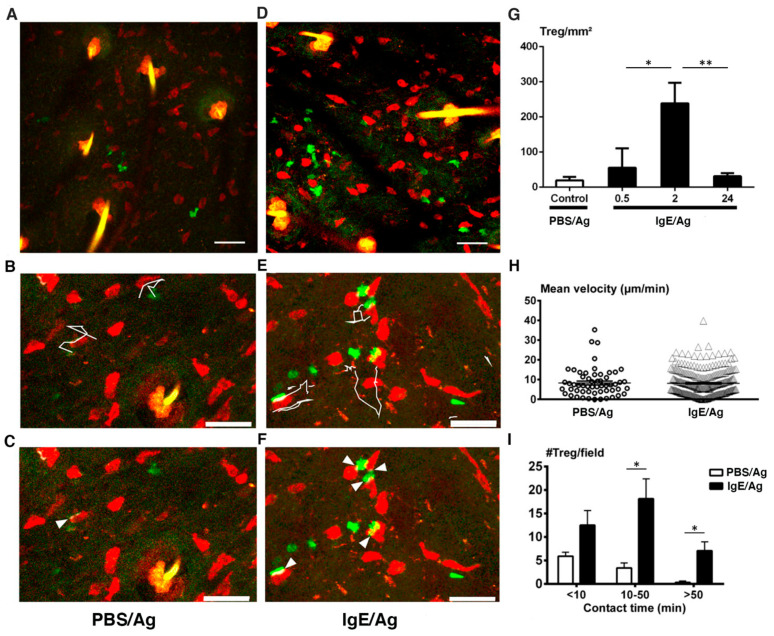
Mast cells interact with Tregs after IgE-mediated degranulation. Photomicrographs showing the distribution (**A**,**D**), in situ mobility (**B**,**E**) and contact profile (**C**,**F**) of Foxp3-eGFP^+^ Tregs as analyzed by intravital confocal microscopy of control PBS-sensitized (**A**–**C**) or IgE-sensitized (**D**–**F**) ear skin of RMB-Foxp3-eGFP mice 2 to 3 h after challenge with Ag (DNP-HSA). (**G**) Tregs were enumerated in control PBS- or IgE-sensitized ear skin of RMB-Foxp3-eGFP mice after challenge with Ag (DNP-HSA) at the indicated time points. Each quantification was performed in at least two independent fields of skin of over 400 × 400 µm Z projection images from three mice and expressed as the mean ± SEM of the total number of Tregs per mm^2^. (**B**,**C**,**E**,**F**) micrographs of reduced size and resolution in 3D dimensions were recorded with a rapid duty cycle of 40 to 60 sec as detailed in the method section. (**B**,**E**) Tregs were tracked with Imaris software (white lines). (**H**) Average Treg velocities and (**I**) MCs-Tregs contact times were measured at 2 h to 3 h after Ag (DNP-HSA) challenge performed in control PBS-sensitized (PBS/Ag) and IgE-sensitized (IgE/Ag) ear skin. (**C**,**F**) Contacts between MCs and Tregs were automatically identified with dedicated software on the time lapse videos (arrow heads), tracked with Imaris software and classified (**I**) into three categories: none or short-lived contacts (0–10 min), intermediate-lived contacts (10–50 min) and sustained long-term contacts (>50 min). Each quantification was performed in at least two independent fields of the ear skin of over 400 × 400 µm Z projection images from three mice. Numbers correspond to the mean ± SEM of Tregs per category. Statistical analysis in G, H and I was based on an unpaired *t*-test between groups. (** p* < 0.05; *** p* < 0.01) Scale bars (**A**,**F**) 50 µm, (**B**–**E**) 30 µm.

**Figure 5 cells-11-03055-f005:**
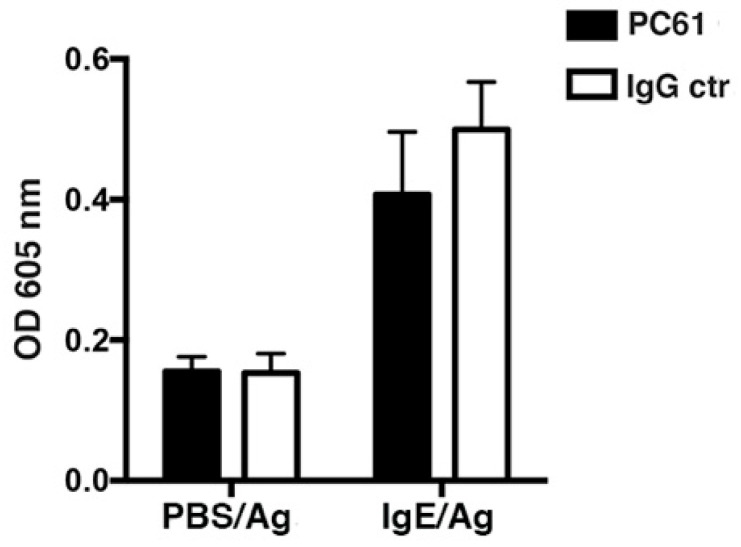
Tregs functional depletion does not affect the acute PCA response RMB mice (n = 6) were treated with rat anti-CD25 (PC61) or irrelevant IgG (IgG ctr) Ab (0.2 mg, ip) to deplete Tregs or not at day 0, before passive sensitization by subcutaneous (i.d.) ear injections at day 2 with anti-DNP IgE (IgE/Ag) or control PBS (PBS/Ag). At day 4, all mice were challenged i.v. with Ag (DNP-HSA 150 μg) in 1% Evans blue (final volume 150 µL) and were then sacrificed 30 min later to determine Evans blue extravasation. Quantification of Evans blue extracted from ears of mice (n = 6) in corresponding experiments is shown. No significant differences between the PC61-treated and IgG control-treated mice were noticed for ears sensitized with IgE (IgE/Ag) or with PBS (PBS/Ag) (unpaired *t*-test between groups).

**Figure 6 cells-11-03055-f006:**
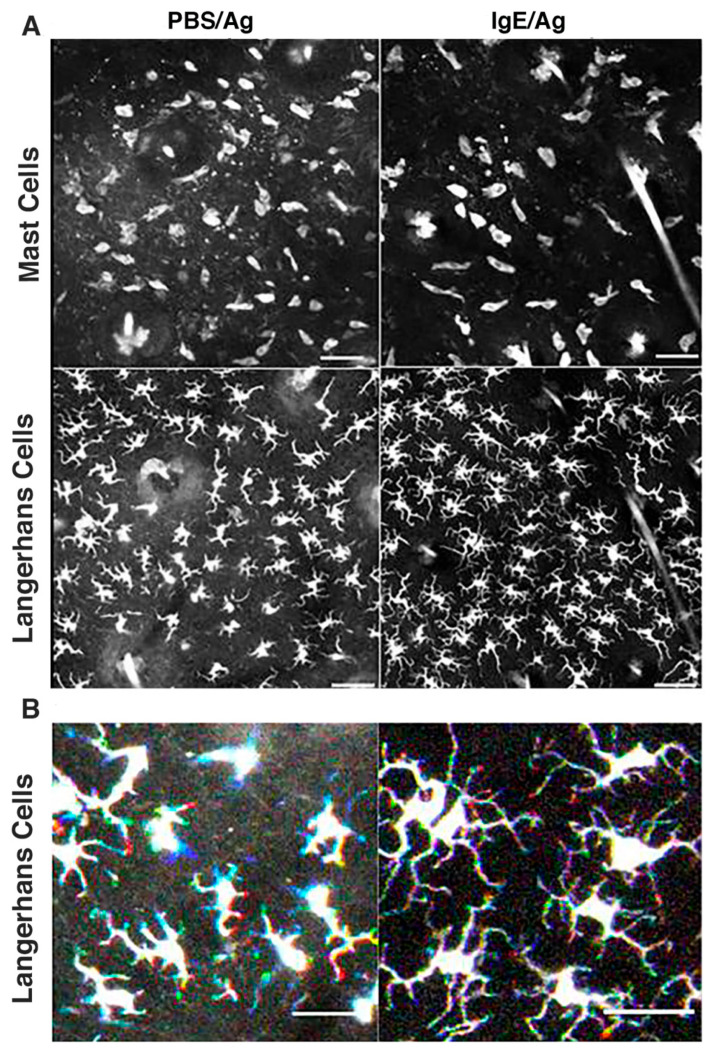
LCs are sessile and exhibit constant sprouting before and after PCA. RMB mice were crossed with knock-in Langerin-eGFP mice and subjected to intravital confocal imaging of ear skin to unravel motility of Langerin-eGFP^+^ LCs in control PBS-sensitized (PBS/Ag) and IgE-sensitized (IgE/Ag) ear skin 3 h after Ag challenge. Dual fluorescence 4D data sets with Z thickness of 45 µm and 3 µm Z step size are processed as for dual channel Foxp3-eGFP^+^ Treg/tdT^+^ MC data sets of Figure 4. (**A**) compares general appearance of Langerin-eGFP^+^ LCs and tdT^+^ MCs. (**B**) To visualize mobility of Langerin-eGFP^+^ LCs, the merge of three sequential time points (0 min, 7 min and 14 min) of an RGB image is represented as in Figure 1A. LCs are sessile and cell body appears in white, while their dendrites emerge with red or blue colored extremities in RGB mode both in control and PCA subjected ears. Scale bar (**A**) 50 µm and (**B**) 30 µm. Typical micrographs were extracted from a pool of experiments performed in three mice.

## Data Availability

All data are contained within the article or Appendix A.

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
