# Peer review of "Mast Cell Interaction with Foxp3+ Regulatory T Cells Occur in the Dermis after Initiation of IgE-Mediated Cutaneous Anaphylaxis"

_cells, 2022, doi:10.3390/cells11193055_

Round 1
Reviewer 1 Report
The paper is interesting because you show a new Knowledge of the role of mast cells in the skin inflammation. Good the new method to observe and calcolate the number of mast cells.
Line 75: delete and, you write it twice
Line 102: delete passive, you weite it twice
Author Response
In response to reviewer 1
Comment: The paper is interesting because you show a new Knowledge of the role of mast cells in the skin inflammation. Good the new method to observe and calcolate the number of mast cells.
Answer: We thank the reviewer for his/her positive comments finding our manuscript interesting and providing new knowledge.
Comment: Line 75: delete and, you write it twice
Answer: We have now corrected this misspelling
Comment: Line 102: delete passive, you weite it twice
Answer: We have now corrected this misspelling
Reviewer 2 Report
Classification and analysis of contacts between MC and Treg cells with different durations provides new fundamental data on the involvement of MCs in innate and adaptive immunity responses.
However, the reviewer has a number of questions.
1. The authors characterize mast cells as "MC are sessile". What mast cells are discussed in the article: tryptase-positive, chymase-positive, or carboxypeptidase A3-positive?
2. The second question is a continuation of the first. How can the authors claim that their microphotographs represent cells, and not fragments of the cytoplasm or post-cellular structures? In particular, there are enough data in the literature that indicate the possibility of mast cells to lose the nucleus, while the remaining cytoplasm remains in the tissue and continues to function for some time. See, e.g., Int. J. Mol. Sci. 2022, 23, 8930. https://doi.org/10.3390/ijms23168930.
3. Questions for Figure 1
3.1. Can the authors claim that all cells shown in Figure 1 have nuclei?
3.2 . What do the authors attribute to the large morphometric characteristics of mast cells in Figure 1A? Judging by the scale, the size of mast cells in vivo should be about 40 microns.
3.3. Why is the mast cell size smaller in Figure 1b compared to Figure 1a.
6.2. In Figures 1B and 1E, mast cells should be indicated with arrows.
6.3. In figures 1D and 1C, the arrows are not explained in the legend.
6.4. In the reviewer's opinion, the number of mast cells in Figures 1a and 1b is not comparable.
6.5. By what mechanism can the authors explain the strong decrease in the number of mast cells by the 6th day of the experiment?
4. To what extent is it possible to transfer the Tandem dimer Tomato (TdT) label from mast cells to other cells through various mechanisms?
5. Questions for Figure 3
5.1. The authors write: “Tregs are motile (Video S1) and occasionally can be found to interact with MC (Figure 3A arrows). How can the authors explain the different spectral characteristics of cells colocalized with mast cells?
5.2. In Figure 3A, authors should indicate Foxp3eGFP+ Tregs with an arrow.
6. Did the authors check the correspondence of the results obtained in the article with the data from the study of biopsy material using multiple immunolabeling?
7. In the caption to Figure 4, the phrase "Scale bars: 50μm" is indicated twice. Should be left only at the end of the legend.
8. In the legend for Figure 4, it is necessary to correct the label (C,E) to (C,F)
9. In the legends for Figure 4, it is difficult to understand the belonging of microphotographs on the left (A, B, C) and on the right (D, E, F).
10. How can the authors explain "constant sprouting before and after PCA"? How can processes of dendritic cells change their histotopographic position in the epidermis so quickly (within a few minutes)?
11. The missprint "homeostasis nor after MC degranulation" on line 449.
Author Response
In response to reviewer 2
Comment: Classification and analysis of contacts between MC and Tregs cells with different durations provides new fundamental data on the involvement of MCs in innate and adaptive immunity responses.
Answer: We thank the reviewer for his/her positive comments finding that our manuscript provides new fundamental knowledge on the classification and analysis of the interaction between MC and Tregs. We also thank the reviewer for all the comments and fruitfull suggestions relative to the MS.
Question 1: The authors characterize mast cells as "MC are sessile". What mast cells are discussed in the article: tryptase-positive, chymase-positive, or carboxypeptidase A3-positive?
Answer: Concerning the remark on sessile MC, we do not claim that MC are completely sessile. Indeed, the very fact that they have to infiltrate the tissues makes clear that they are able to move. However, this probably takes place in a much longer time scale as the one followed here (min to 2/3hrs). Concerning the protease staining, the tdT fluorescence is driven by the FceRI b chain promoter with the b chain being required for a functional expression of FceRI that is expressed in all tissue mast cells. Therefore, in principle, all tissue mast cells e.g. mucosal (tryptase+) and serosal (chymase+, CPA3+) mast cells are supposed to get stained. In agreement, in addition to published data showing extensive staining of skin mast cells (Ngo Nyekel et al, Frontiers Immunol, 2018), MC extracted from skin, peritoneal MC and immature bone marrow derived MC (Dahdah et al, J Clin Invest. 2014;124(10):4577-4589), ankle joint mast cells (van der Velden et al. Arthritis Research & Therapy (2016) 18:138 ) as well as unpublished flow cytometric analysis of gut extracted mast cells, lung extracted MC, liver-extracted MC stain all of which stain positive for tdT. However the reviewer is right, that a systematic analysis needs to be performed to visualize whether FceRI-negative differentiated MC exist in tissue.
To bring this point to the readers, we have now added a comment to the methods section paragraph entitled “Mouse strains” to clearly describe the pattern of expression of tdT and diphtheria toxin receptor in RMB mice.
Question 2 and Question 3.1:
- The second question is a continuation of the first. How can the authors claim that their microphotographs represent cells, and not fragments of the cytoplasm or post-cellular structures? In particular, there are enough data in the literature that indicate the possibility of mast cells to lose the nucleus, while the remaining cytoplasm remains in the tissue and continues to function for some time. See, e.g., Int. J. Mol. Sci. 2022, 23, 8930. https://doi.org/10.3390/ijms23168930.
3.1: Can the authors claim that all cells shown in Figure 1 have nuclei?
Answer: We thank the referee for this important question relevant for all our work. We have now improved the quality of Figure 1 realizing that the resolution was altered during its processing needed to produce the final pdf document. We can now clearly visualize in Figure 1A that most MCs have a nucleus hole corresponding to the fact that the nuclei exclude the cytoplasmic tdT dye. We also double checked the quality of all final figures. Of note, in further micrographs starting at figure 1B, the maximum intensity projection software, which is needed to project the 3D MCs on 2D images erases most nuclei since they are surrrounded by tdT staining on all sections of the cells. Fewer objects of small size can also be present on our micrographs but they are clearly different from the characteristic elongated shape of MCs with a usual size of about 20 to 30 micron or sometimes even longer when highly elongated. Although enucleated cells, cell fragments, post-cellular structures exist, this is probably more the exception than the rule (eventually being reinforced during an inflammatory/cancerous process as shown in the cited publication of the reviewer). Based on our own analysis assembling the photographs as well as the published cytofluorometric analysis of mast cells extracted from skin of RMB mice (please see the initial publication, Dahdah et al, J Clin Invest. 2014;124(10):4577-4589) the large majority (> 98%) behaves as real cells even after the harsh treatment of cell extraction. Likewise we are enclined to think that our micrographs of Figure 1 and further Figures essentially show tdT-positive mast cells.
To bring this point to the readers, we have now rephrased a whole paragraph entitled Static mosaic images and time lapse video acquisitions in the methods section of the MS.
Question 3.2: What do the authors attribute to the large morphometric characteristics of mast cells in Figure 1A? Judging by the scale, the size of mast cells in vivo should be about 40 microns.
Answer: The reviewer is right, their was an error regarding the size bar in Figure 1A as it should be 30 microns not 50 as indicated. This is now corrected
Question 3.3: Why is the mast cell size smaller in Figure 1b compared to Figure 1a.
Answer: Following our correction indicated above, the mast cells are now of similar size
Question 6.2: 6.2. In Figures 1B and 1E, mast cells should be indicated with arrows.
Answer: as requested MCs are now indicated by arrows also in Figure 1B and IE
Question 6.3: In figures 1D and 1C, the arrows are not explained in the legend.
Answer: as requested we now explain all arrows of Figure 1 in the legend.
Question 6.4: In the reviewer's opinion, the number of mast cells in Figures 1a and 1b is not comparable.
Answer: This impression is correct and is explained by the fact that Figure 1A is a single confocal section and other MCs may reside above or below this plane in the dermis. In Figure 1B to E, we applied a maximum projection software operating on the whole 3D volume, which necessarily explores all MCs in the dermis. Hence this true impression of having more MCs, at the expense of losing some details such as the nuclear holes was required to quantify the number of MCs per mm2 during the time course of the depletion/repletion kinetics shown in Figure 1B to F. Moreover, all scale bars have been checked, leading to some corrections in the legends to figures (see above).
We have now added a comment in the result section of the MS in first paragraph entitled “Mast cells are sessile and, after conditional ablation, undergo slow repopulation kinetics in the skin”
Question 6.5: By what mechanism can the authors explain the strong decrease in the number of mast cells by the 6th day of the experiment?
Answer: the strong decrease we are observing is due to the treatment with diphteria toxin, which induces apoptotic cell death of MCs upon i.p. injection. Indeed, MCs in RMB mice besides expressing tdT also express the diphteria toxin receptor and therefore can be deleted by treatment with diphteria toxin (see M&M sections as well as the original article reporting RMB mice: Dahdah et al, J Clin Invest. 2014;124(10):4577-4589). To make this clearer we now have explained this better in the legend to figure 1B to E.
Question 4. To what extent is it possible to transfer the Tandem dimer Tomato (TdT) label from mast cells to other cells through various mechanisms?
Answer: Transfer of molecules from cell to cell has clearly been reported involving for example trogocytosis, exosome-related mechanisms, gap junctions, plasmodesmata tuneling nanotubes etc. In addition, literature on MC has shown that after activation through the IgE receptor they can form conjugates via integrins with dendritic cells (DC) and thereby transfer material to the DCs likely involving an exosome related mechanism ( J. Cell Biol. Vol. 210 No. 5 851–864). Under these conditions the transferred material exhibits a punctuate cellular staining. Therefore, the possibility of exchange of tdT cannot completely be excluded. Yet, in this regard previous analysis (Dahdah et al, J Clin Invest. 2014;124(10):4577-4589) has shown that essentially all tdT-positive MC cells isolated from the peritoneal cavity, carry a MCsignature, while it is true that when cells are extracted from skin some tdT-positive cells do not carry the MC-specific marker. However, this concerns only a minor proportion of cells (<2%) and, besides possible intercellular transfer, could also be due to the somewhat harsh extraction procedure before cytofluorometric analysis. Therefore, although examining the phenomenon of intercellular transfer remains an interesting topic, we believe that in our context essentially all imaged MCs are indeed MCs. Of note, the fact that at day 6 post depletion there is a drastic decrease of MCs in the dermis indicates that the tdT initially present in MCs has been fully processed and eliminated. It is also clear that the fluorescent property of tdT is necessarily quite fragile and may not last more than a few hours if transferred passively to neighboring cells.
Question 5.1: The authors write: “Tregs are motile (Video S1) and occasionally can be found to interact with MC (Figure 3A arrows). How can the authors explain the different spectral characteristics of cells colocalized with mast cells?
Answer: We thank you for this point, we have reformulated the results to clarify that tdT+ MCs and eGFP+Tregs have different fluorescence characteristics. We have changed the order of the description and more clearly indicated that only the Tregs exhibit a motile behavior.
Cf the result section in the paragraph entitled “Mast cells interact with Tregs after initiation of the PCA”
Question 5.2: In Figure 3A, authors should indicate Foxp3eGFP+ Tregs with an arrow.
Answer: As requested we have now indicated Foxp3eGFP+ Tregs in Figures 3A and C
Question 6: Did the authors check the correspondence of the results obtained in the article with the data from the study of biopsy material using multiple immunolabeling?
Answer: if the reviewer refers to the MC counts obtained by biopsy, our own toluidine blue MC counts in ear skin (Scandiuzzi et al, J. Immunol, 2010) yield about 100 MC/mm2, which is about 3 to 4 times less that the numbers obtained here. This is due to the fact that our confocal analysis comprises essentially the whole dermis layer, while biopsy slices do not comprise the whole dermal layer.
Question 7: In the caption to Figure 4, the phrase "Scale bars: 50μm" is indicated twice. Should be left only at the end of the legend.
Answer: We have corrected this error
Question 8: In the legend for Figure 4, it is necessary to correct the label (C,E) to (C,F)
Answer: as requested we have now corrected this error.
Question 9: In the legends for Figure 4, it is difficult to understand the belonging of microphotographs on the left (A, B, C) and on the right (D, E, F).
Answer: We agree with the reviewer and we have now added at the bottom of the left and right panels the experimental conditions, namely after a challenge with PBS (left panels) or after challenge with the Ag (DNP-HSA) (right panels).
Question 10: How can the authors explain "constant sprouting before and after PCA"? How can processes of dendritic cells change their histotopographic position in the epidermis so quickly (within a few minutes)?
Answer: In Figure 6B, we have superimposed 3 time points from a 4D video confocal imaging. Basically, each 3D time frame is projected (maximum intensity projection) and 3 time frames are selected at time 0, 7 and 14 min coded in red, green, blue respectively. We see that although the LC cell bodies are immobile, the extremities of LC dendrites are either white (no motion), red, blue or green, testifying small movements of their extremities within a typical time scale of about 7 min.
We thank the referee and agree that the result section was not explicit enough. We have therefore reformulated this in the paragraph entitled “Langerhans cells remain sessile while exhibiting constant sprouting of their dendrites”
Question 11: The missprint "homeostasis nor after MC degranulation" on line 449.
Answer: we have corrected this error
Reviewer 3 Report
The aim of this study was to show the interaction of MCs, Treg cells and LC cells in IgE-dependent passive skin allergy responses. Mast cells are known to have an important role in IgE-mediated skin allergy and it is therefore of great interest to investigate the mechanism of their role in skin allergy. This paper is interesting and could make a good contribution to the scientific literature. However, there are some minor details in the text that need to be revised.
1. Line 19-23, What does "MC-dependent" mean? Is it a writing error?
2. Line 41, here IgE appears for the first time in the text and should be given its full name. cCD1, cCD2, cCD3 and IL-9 etc. have the same problem(Line 48 & 63). Please double-check throughout the text.
3. Line 50, the full name of "Treg" is different from the abstract, please check carefully.
4. Line 50, “Foxp3+” should be “Foxp3+”. Please revise in full paper.
5. Line 59-60, is there a mistake in the full name of ILC2? Please confirm again.
6. Line 67-68, This is missing punctuation.
7. Line 84-99, How many mice were used in the experiment? How were they grouped? Please specify in the protocol.
8. Line 88, there is an extra space between "MC" and "and".
9. Lines 106,115 and 257 etc. The unit of milliliter in the text is either "µl" or "µL", so please standardize the format throughout the text.
10. Line 144, Please provide details in theplan as to why the monitoring time is 30-60 minutes.
11. Line 204, “E)” is an error. Line 281 and 286, With the same problem, please check in full.
12. Line 272, what’s the means of “Figure 3A arrows”?
13. Line 289-304, For the use of "Figure", please refer to the full text.
14. Line 322, The "P" in (*P < 0.05; **P < 0.01) should be in italics.
15. Line 359, Fi(A), incorrectly written.
16. Line 389, An extra preposition.
17. Line 406, “TGFβ” should be “TGF-β”.
18. Line 422, what is the means of “Treg velocity”?
19. Treg" and "Tregs" appear several times across the text, but "Tregs" does not have a full name the first time it appears, perhaps an error in the author's writing? And Treg is simply an abbreviation for "regulatory T", not "regulatory T cells."
20. Figure 5, The figure size needs to be resized.
Author Response
In response to reviewer 3
Comments and Suggestions for Authors
Comment: The aim of this study was to show the interaction of MCs, Treg cells and LC cells in IgE-dependent passive skin allergy responses. Mast cells are known to have an important role in IgE-mediated skin allergy and it is therefore of great interest to investigate the mechanism of their role in skin allergy. This paper is interesting and could make a good contribution to the scientific literature. However, there are some minor details in the text that need to be revised.
Answer: We thank the reviewer for his/her positive comments on the manuscript and for all the comments and fruitfull suggestions relative to the MS.
Comment: Line 19-23, What does "MC-dependent" mean? Is it a writing error?
Answer: this was to indicate that the PCA requires MC activation. We agree with the reviewer that the term is somewhat misleading and have either deleted the term (abstract) or replaced by MC-induced.
Comment: Line 41, here IgE appears for the first time in the text and should be given its full name. cCD1, cCD2, cCD3 and IL-9 etc. have the same problem(Line 48 & 63). Please double-check throughout the text.
Answer: as requested we have spelled out all abbreviations in the text when appearing for the first time.
Comment: Line 50, the full name of "Treg" is different from the abstract, please check carefully.
Answer: as requested we have now unified the names. Treg is used as an abbreviation for regulatory T lymphocyte, LC as an abbreviation for Langerhans cell and MC as an abbreviation for mast cell. When the term is used in pluriel, we added and s to the abbreviation which becomes Tregs, LCs and MCs.
Comment: Line 50, “Foxp3+” should be “Foxp3+”. Please revise in full paper.
Answer: as requested we have revised to Foxp3+ in the full paper.
Comment: Line 59-60, is there a mistake in the full name of ILC2? Please confirm again.
Answer: the term is correct, but we have deleted the term dermal and added it after. It now reads: “type 2 innate lymphoid cells (ILC2) in the dermis”
Comment: Line 67-68, This is missing punctuation.
Answer: We have changed the punctuation and phrasing as requested.
Comment: Line 84-99, How many mice were used in the experiment? How were they grouped? Please specify in the protocol.
Answer: The number of mice was indicated in the legends for all figures except Figure 6. We have now corrected this error.
Comment: Line 88, there is an extra space between "MC" and "and".
Answer: We have corrected this error
Comment: Lines 106,115 and 257 etc. The unit of milliliter in the text is either "µl" or "µL", so please standardize the format throughout the text.
Answer: We have unified to µL
Comment: Line 144, Please provide details in theplan as to why the monitoring time is 30-60 minutes.
Answer: We have now completely reformulated the relevant paragraph in the methods section entitled Static mosaic images and time lapse video acquisitions to better explain the monitoring time.
Comment: Line 204, “E)” is an error. Line 281 and 286, With the same problem, please check in full.
Answer: We have now revised the manuscript and corrected these errors.
Comment: Line 272, what’s the means of “Figure 3A arrows”?
Answer: The meaning of the arrows is now more clearly explained in the legend to the figure
Comment: Line 289-304, For the use of "Figure", please refer to the full text.
Answer: We corrected this error.
Comment: Line 322, The "P" in (*P < 0.05; **P < 0.01) should be in italics.
Answer: We have corrected this error as requested
Comment: Line 359, Fi(A), incorrectly written.
Answer: We corrected this error.
Comment: 16. Line 389, An extra preposition.
Answer: We corrected this error.
Comment: Line 406, “TGFβ” should be “TGF-β”.
Answer: We corrected this error.
Comment: Line 422, what is the means of “Treg velocity”?
Answer: The Treg velocity is calculated in micron per min from image stack data analysis as precised now in the Methods section.
Comment: Treg" and "Tregs" appear several times across the text, but "Tregs" does not have a full name the first time it appears, perhaps an error in the author's writing? And Treg is simply an abbreviation for "regulatory T", not "regulatory T cells."
Answer: As already stated above for your comment related to line 50 we have unified the whole MS according to the definition of Treg and Tregs.
Comment: Figure 5, The figure size needs to be resized.
Answer: We have resized Figure 5 in the revised version.